# New Insights in Autoimmune Hemolytic Anemia: From Pathogenesis to Therapy

**DOI:** 10.3390/jcm9123859

**Published:** 2020-11-27

**Authors:** Wilma Barcellini, Anna Zaninoni, Juri Alessandro Giannotta, Bruno Fattizzo

**Affiliations:** Fondazione IRCCS Ca’ Granda Ospedale Maggiore Policlinico, University of Milan, 20100 Milan, Italy; anna.zaninoni@policlinico.mi.it (A.Z.); jurigiann@gmail.com (J.A.G.); bruno.fattizzo@unimi.it (B.F.)

**Keywords:** warm autoimmune hemolytic anemia, cold agglutinin disease, complement, direct antiglobulin test, cytokines, therapies

## Abstract

Autoimmune hemolytic anemia (AIHA) is a highly heterogeneous disease due to increased destruction of autologous erythrocytes by autoantibodies with or without complement involvement. Other pathogenic mechanisms include hyper-activation of cellular immune effectors, cytokine dysregulation, and ineffective marrow compensation. AIHAs may be primary or associated with lymphoproliferative and autoimmune diseases, infections, immunodeficiencies, solid tumors, transplants, and drugs. The direct antiglobulin test is the cornerstone of diagnosis, allowing the distinction into warm forms (wAIHA), cold agglutinin disease (CAD), and other more rare forms. The immunologic mechanisms responsible for erythrocyte destruction in the various AIHAs are different and therefore therapy is quite dissimilar. In wAIHA, steroids represent first line therapy, followed by rituximab and splenectomy. Conventional immunosuppressive drugs (azathioprine, cyclophosphamide, cyclosporine) are now considered the third line. In CAD, steroids are useful only at high/unacceptable doses and splenectomy is uneffective. Rituximab is advised in first line therapy, followed by rituximab plus bendamustine and bortezomib. Several new drugs are under development including B-cell directed therapies (ibrutinib, venetoclax, parsaclisib) and inhibitors of complement (sutimlimab, pegcetacoplan), spleen tyrosine kinases (fostamatinib), or neonatal Fc receptor. Here, a comprehensive review of the main clinical characteristics, diagnosis, and pathogenic mechanisms of AIHA are provided, along with classic and new therapeutic approaches.

## 1. Introduction

Autoimmune hemolytic anemia (AIHA) is a highly heterogeneous disease from fully compensated to life-threatening, basically due to increased destruction of autologous red blood cells (RBC) by several immune mechanisms. The key players are autoantibodies with or without complement (C) involvement; however, several cellular immune effectors are increasingly recognized, along with cytokine dysregulation and ineffective bone marrow compensation [1,2]. AIHAs may be primary (idiopathic) or associated with several conditions (lymphoproliferative, autoimmune and infectious diseases, immunodeficiencies, solid tumors, transplants, and drugs) where the cited immunologic mechanisms are variably combined [3,4,5,6]. Diagnosis may be easy, but some rare forms may be challenging, causing harmful delay in appropriate therapy [7]. Based on the isotype and thermal characteristics of the autoantibody, AIHAs have been traditionally divided into warm forms, cold agglutinin disease (CAD), and more rare forms (mixed, atypical and paroxysmal cold hemoglobinuria, PCH). This classification is fundamental as the immunologic mechanisms responsible are dissimilar and consequently therapy may be different. Therapy of AIHA is mainly based on expert opinions [4,5,8] and few prospective trials [9,10,11,12,13,14,15]. However, new targeted drugs are increasingly appearing, providing the possibility of harnessing therapy based on the main immunologic mechanisms involved [4,7,8,16]. Here, we provide a comprehensive review of the main clinical characteristics, diagnosis, and pathogenic mechanisms, along with classic and new therapeutic approaches to the disease.

## 2. Epidemiology

AIHA is a relatively rare disease (estimated incidence of 1–3 cases per 100,000/year) occurring more commonly after the age of 40 years, but also in early childhood. In the latter, disease is generally acute and transient as it is mainly associated with viral infections. In adults, chronic/relapsing cases are more frequent, particularly if associated with autoimmune or lymphoproliferative disorders. Regarding prognosis, survival in adults is reported to be 91% at one year, 75% at five years, and 73% at 10 years. Mortality is reported to be 4%, mainly due to infections, simultaneous autoimmune thrombocytopenia (Evans syndrome), acute renal failure, and major thrombotic events [1,4,17,18]. In a recent large multicenter study of patients with CAD, the prevalence of the disease was 4-fold different between cold and warmer climates (20 versus 5 cases/million) as well as the incidence (1.9 versus 0.48 cases/million per year) [19].

## 3. Classification and Diagnosis

AIHAs can be classified in several ways based on the thermal characteristics of the autoantibody, on the autoantibody isotype, and on the acute versus chronic/relapsing clinical presentation. The first two criteria are the most useful and are based on the direct antiglobulin test (DAT) or Coombs test, and on the indirect antiglobulin test (IAT) or indirect Coombs test. The first demonstrates the presence of autoantibodies on the patient’s RBCs (Red blood cells) and defines their class, thermal characteristics, and ability to activate complement, and the second that allows their identification is serum. The latter may be completed by the study of the eluate (i.e., the antibodies eluted by the DAT-positive RBCs), which may be useful in difficult cases [1,7,8,20]. Table 1 shows the traditional AIHA classifications. Warm AIHAs (wAIHAs) represent the majority of all forms (60–70%) and are generally due to an IgG that binds to RBCs at a temperature around 37 °C. The DAT is typically positive with anti-IgG antisera or IgG plus C at low titer. The second more frequent AIHA is cold agglutinin disease (CAD) in 20–25% of cases, which is due to an IgM autoantibody that has an optimum temperature of reaction at 4 °C (thermal range 4–34 °C) and strongly activates complement. The DAT is positive with anti-C antisera and a high titer of cold agglutinins is found in the serum. Typically, a spontaneous agglutination of erythrocytes occurs at room temperature, invalidating automated blood counts and raising the diagnostic suspect. Finally, the mixed forms (5–10% of patients) show common characteristics of wAIHA and CAD, with a DAT positive for both IgG and C and high titer cold agglutinins. As an exception, PCH (1–5% of cases) is due to an IgG that binds to RBCs in the cold, but causes severe intravascular hemolysis at 37 °C and is diagnosed with the Donath–Landsteiner test [1,20]. Finally, AIHA may be classified as primary or secondary according to the absence/presence of underlying conditions. These include infections, lymphoproliferative syndromes, other autoimmune disorders, congenital immunodeficiencies, and others [4,6]. This distinction is important since it may impact on therapy choice, as discussed later. Regarding cold forms, the term CAD refers to cold antibodies secondary to any clonal lymphoproliferative disorder, whilst cold agglutinin secondary to infectious or other malignant disease should be referred to as cold agglutinin syndrome (CAS) [4].

## 4. Difficulties in Autoimmune Hemolytic Anemia (AIHA) Diagnosis

There is increasing awareness of the atypical forms that include IgA driven, DAT negative, and warm IgM AIHAs. The first cases are due to an IgA antibody, which is frequently, but not always associated with an IgG. This AIHA is difficult to diagnose without the use of anti-IgA antisera, which is not routine in most laboratories. In fact, the DAT can be performed with various methods, the most classic being a test tube with polyspecific and monospecific antisera (anti-IgG, anti-C, anti-IgA and anti-IgM). More sensitive tests, able to reveal smaller amounts of antibodies bound to RBCs, include microcolumn and solid phase tests, now widely used in routine diagnosis. More sophisticated methods comprise washing RBCs with low ionic strength solutions (LISS) that are able to detect low affinity autoantibodies, and experimental tests such as immunoradiometric assays, ELISA, and cytometry. Concerning the sensitivity of the various methods, the classic DAT-tube effectively diagnoses AIHA when at least 500 autoantibody molecules are bound to RBCs; the microcolumn test requires about 200–300 molecules/RBC; and cytometry, the most sensitive technique, is able to detect even 30–40 autoantibody molecules per RBC [2,4,21]. Despite the use of all the methods described above, 5–10% of AIHAs remain DAT-negative [1,17]. In these cases, the diagnosis is made on exclusion and on the basis of the clinical response to steroid therapy. Finally, it is worth a reminder that AIHAs due to IgM with a thermal range close to body temperature (warm IgM) are fortunately rare, but able to strongly activate C in vivo and cause massive intravascular hemolysis. They can appear weakly DAT positive for C, or DAT negative, possibly delaying diagnosis and proper therapy, and thus the results are extremely serious and potentially fatal. A useful diagnostic tool is the DDAT (Dual Direct Antiglobulin Test) [22].

Notably, the DAT may be falsely positive after the administration of various therapies (intravenous immunoglobulins, Rh immunoglobulins, and anti-thymocyte globulins) and in diseases with elevated serum gammaglobulins or paraproteins, in the presence of IgG/IgM anti-cardiolipin antibodies and lupus anticoagulant, and seldom in mechanical hemolytic anemia. Recently, a false DAT positivity has been reported after daratumumab, an anti-CD38 antibody for the treatment of multiple myeloma, since CD38 is also expressed on RBCs [23]. Moreover, the DAT may be positive for the presence of alloantibodies in patients who have recently been transfused, during delayed hemolytic transfusion reactions, and in the hemolytic disease of the newborn. Finally, the DAT may be positive without clinical evidence of AIHA in a small percentage of healthy blood donors (<0.1%) and hospitalized patients (0.3–8%) [1].

The diagnosis of DAT-negative AIHAs often represents a critical problem, particularly in emergency situations. In fact, all of the numerous causes of hemolysis should be considered, which include congenital anemias due to defect in membrane proteins (spherocytosis, ellisocytosis, stomatocytosis), defects in erythrocytic enzymes (glucose 6 phosphate dehydrogenase, pyruvate kinase, and others more rare), paroxysmal nocturnal hemoglobinuria, mechanical hemolysis (walking hemoglobinuria, heart valve prosthesis, severe aortic stenosis, arteriovenous malformations), disseminated intravascular coagulation, hemolytic uremic syndrome, eclampsia, thermal insults (extensive burns), serious infections (malaria, Clostridium sepsis, Babesia, Bartonella), poisons (snakes, spiders, acute copper poisoning, Wilson’s disease) as well as numerous drugs [4,7,24].

## 5. Other Investigations in AIHA

Although not generally performed in AIHA at onset, bone marrow evaluation (morphology, cytometry, cytogenetics, and biopsy) has been recently advised in CAD at diagnosis and in wAIHA relapsed after or refractory to first-line therapy with steroids [4,7,16]. This recommendation is based on the importance of diagnosing secondary forms to lymphoproliferative disease and to evaluate bone marrow compensation, particularly in borderline cases of myelodysplasia or bone marrow failure. Moreover, bone marrow biopsy may reveal the type of lymphocyte infiltrate (T or B) or the presence of fibrosis/dyserythropoiesis, possibly guiding a better target-therapy and avoiding excessive immunosuppression. Furthermore, the recommended diagnostic workup includes whole-body CT scan, and serology for infections and autoimmune diseases including anti-cardiolipin and beta-2 glycoprotein 1. The determination of endogenous erythropoietin levels is advised in relapsed/refractory and heavily treated subjects, since these patients may benefit from recombinant erythropoietin administration [25]. Finally, molecular analysis and next-generation sequencing would help to confirm associated conditions in difficult cases (congenital anemias, immunodeficiencies, myelo and lymphoproliferative disorders).

## 6. Pathogenic Mechanisms of RBC (Red Blood Cell) Destruction

The definition of the type and thermal characteristics of the autoantibody is of fundamental importance since it determines different mechanisms of hemolysis, with a consequently different clinical picture and therapeutic approach [1,2]. IgG autoantibodies are monomers that weakly fix the complement system; they cause RBC destruction via the antibody-dependent cellular cytotoxicity (ADCC), driven by cells of the monocyte-macrophage system that phagocyte RBCs through the Fc fragment of IgG (or the complement fractions C3b). Activated lymphocytes that express receptors for the IgG Fc fragment and for C3b may also mediate an ADCC. Therefore, hemolysis is extravascular and occurs mostly in the spleen in the case of macrophage-mediated ADCC, and in the liver in the case of C3b-mediated ADCC. Notably, the spleen represents the hemocatheretic organ that removes all damaged/senescent cells of the blood, but it is also a lymphatic organ, and therefore able to contribute to autoantibody production. At variance with IgG, IgM autoantibodies are pentamers with high avidity and ability to activate the complement cascade until the final lytic complex (C5–C9). The latter determines the lysis of RBCs directly in the circulation (intravascular hemolysis) through the activation of “perforins” and other cytotoxic factors. It has been calculated that extravascular hemolysis causes the destruction of approximately 0.25 mL of RBCs per kg of body weight per hour (approximately 420 mL of RBCs in 24 h for a 70 kg patient); at variance, IgM-mediated intravascular hemolysis causes the destruction of about 200 mL of RBCs per hour, with a consequent greater clinical severity (Figure 1).

Finally, an important player in determining the pathogenesis and the clinical severity of AIHA is the hematopoietic bone marrow, which compensates anemia by increasing erythropoiesis. The marker of this phenomenon is the increased number of reticulocytes (immature RBCs with nuclear residue) in the peripheral blood. Reticulocytosis is in fact a “good” sign in the course of AIHA, indicating efficient bone marrow compensation. Conversely, in the case of severe anemia with reticulocytopenia, the prognosis is poor, with greater clinical severity and slower response to therapies [1,17,18]. In some cases, reticulocytopenia may be due to autoimmunity against bone marrow precursors [2,7].

## 7. Genetic Background in AIHA Pathogenesis

The genetic background plays an important role. AIHA has been associated with HLA-B8 and BW6 locus, with a particular configuration of the variable region of the immunoglobulin heavy and light chains (IGHV and IGKV), and with polymorphism of the cytotoxic T-lymphocyte antigen-4 (CTLA-4) gene [26]. More recently, mutations of KMT2D and CARD11 genes have been reported in 69% and 31% of CAD patients [27]. Moreover, mutations in genes implicated in primary immunodeficiencies (TNFRSF6, CTLA4, STAT3, PIK3CD, CBL, ADAR1, LRBA, RAG1, and KRAS) have been reported in about 40% of pediatric Evans syndrome (the association of AIHA and primary thrombocytopenia) [28]. Finally, the close association of autoimmunity and immunodeficiency is highlighted by two immunodeficiencies characterized by increased autoimmune phenomena: the autoimmune lymphoproliferative syndrome (ALPS) and the Kabuki syndrome (KS); the former is marked by organomegalies and by the presence of somatic or germline mutations of genes involved in apoptosis (i.e., FAS, FASL, CASP10, CASP 8, NRAS, or KRAS); KS is marked by malformations and intellectual disability, and is caused by mutations in the KDM6A or KMT2D genes, which are involved in tolerance and immune system maturation [29,30].

## 8. Immunologic Mechanisms of AIHA Pathogenesis

Autoimmunity can arise from several mechanisms, as summarized in Figure 2. One of the most simple is the modification of RBC antigens as a consequence of infectious agents or drugs. During infections, there is the so-called “molecular mimicry”, (i.e., a cross-reaction between antigen determinants of the infectious agent and self molecules). Several infections have been associated with AIHA (parvovirus B19, hepatotropic virus, HIV, mycoplasma pneumonia, mycobacterium tuberculosis, brucella, syphilis) [6,31,32,33,34,35,36] including, more recently, COVID-19 pneumonia [37,38,39]. Drugs can induce the production of autoantibodies by modifying the RBC membrane by adsorption to the membrane or by binding to the membrane as immunocomplexes. In the first case, drug-induced AIHA is indistinguishable from classic primary forms with a DAT positive for IgG and extravascular hemolysis (typical example alpha-methyldopa). In the second case, the autoantibody (generally an IgG) reacts with the drug that is firmly linked to the RBC membrane (typical examples are penicillin, ampicillin, methicillin, carbenicillin, and cephalosporins). In the third case (immunecomplexes of drug-antibody to drug), the antibody is generally an IgM, the hemolysis is intravascular, and the DAT is positive for complement (many drugs involved such as quinidine, phenacetin, hydrochlorothiazide, rifampicin, sulfonamides, isoniazid, quinine, tetracyclines, hydralazine, chloropromazine, and streptomycin [1,24,40,41].

An additional mechanism for the production of autoantibodies is the so-called “emergence of forbidden clones” during B lymphoproliferative syndromes (such as chronic lymphocytic leukemia and lymphomas). This hypothesis was proposed by Burnet more than 60 years ago, and is based on the persistence of self-reactive clones of lymphocytes that should have been deleted in the fetus, but may reactivate and promote autoimmune reactions [42]. More recent insights regarding the production of autoantibodies in patients with CLL. CLL cells may facilitate autoimmunity by direct antigen presentation, and through the production of nonfunctional T regulatory cells that fuels the imbalance between Th17 cells and T-regs. Neoplastic cells may also produce autoantibodies in some cases, however, these are mainly polyclonal IgG produced by non-malignant self-reactive B-cells. Similar to this mechanism is polyclonal B-cell activation, occurring in different conditions such as infection with EBV, CMV, hepatotropic virus, and HIV. Finally, the basis of autoimmune phenomena is due to the break of “self-tolerance”, which is controlled by complex cellular mechanisms and cytokine patterns not yet fully understood [43]. This is the case of congenital immunodeficiencies, where autoimmunity arising from this complex interplay is also sustained by germinal mutations impairing tolerance maturation [7]. Follicular T-helper cells, which are involved in the initiation and maintenance of immune responses that generate memory B cells and long-lived plasma cells, have been recently shown to be increased in autoimmune cytopenias and to contribute to autoantibodies production [2].

## 9. Cytokine Dysregulation in AIHA

There is also evidence of cytokine dysregulation in AIHA. Among the numerous and sometimes conflicting findings, interleukin (IL)-4, IL-6, and IL-10 have been found elevated in patients versus healthy controls [11,44,45]. This is consistent with a prevalent T-helper (Th) 2 humoral response and an antibody-mediated mechanism of RBC destruction in AIHA. Interferon (IFN)-γ has been reported to be reduced in AIHA patients compared with controls, resulting in decreased inhibition of Th 2 response, and consequently in an amplification of the autoantibody-mediated autoimmune disease. Cellular immunity is also involved with increased activity of cytotoxic CD8+T lymphocytes, natural killer cells, and activated macrophages. Moreover, IL-2 and IL-12, which induce the differentiation of CD4+ naïve T cells into the Th 1 subset, have been found elevated, further boosting cellular immunity [2,44]. In line with this over-activation, transforming growth factor (TGF)-β has been reported as elevated. This pleiotropic cytokine favors the differentiation of the Th 17 subset, which amplifies the pro-inflammatory and autoimmune responses [46]. Finally, lymphocyte subsets able to downregulate autoimmune response such as peripheral CD4+ T-regulatory cells have been reported as reduced in AIHA patients compared with the controls, again favoring autoimmune responses (Figure 3).

## 10. Hematologic Parameters in AIHA

The clinical picture and course of AIHA is highly heterogeneus, from insidious to fulminant and with variable degrees of anemia (Table 2). Reticulocytes are generally increased, indirect hyperbilirubinemia and LDH are moderately elevated, and haptoglobin is reduced. Hemoglobinemia, hemoglobinuria and hemosiderinuria may be present in hyper-acute and massive forms. Mixed and atypical forms were generally more severe and with more hemolytic pattern, but overall, a wide range of all parameters was observed [17,18]. In a recent large multicenter study, CAD patients showed median hemoglobin levels of 9.2 g/dL (range, 4.5–15.3 g/dL), and anemia was slight (>10.0 g/dL) in 36% of the patients, moderate (8.0–10.0 g/dL) in 37%, and severe (<8.0 g/dL) in 27% [19].

Reticulocytes, usually expressed as a percentage, should be more correctly converted in absolute numbers, and refer to hemoglobin/hematocrit values. In fact, adequate reticulocytosis during severe anemia is “physiological”, whilst reticulocytopenia (present in 20% of adults and up to 40% of children) is a poor prognostic index [17,47,48]. Reticulocitopenia may represent a medical emergency, be long-lasting despite therapy, and require very high transfusion need [49]. Notably, reticulocytosis may also be present in other causes of anemia (hemorrhage, high altitude acclimatization, pregnancy), and reticulocitopenia may be favored by other causes such as associated bone marrow disease (lymphoproliferative syndromes, myelodysplasia, aplasia) or other causes (renal failure, iron, vitamin B12, or folic acid deficiency).

Regarding other hemolytic markers, high LDH values are characteristic of intravascular hemolysis and hyperacute forms. LDH may be overestimated in the case of tissue necrosis or high turnover (myocardial infarction, pulmonary embolism, acute hepatitis, solid or hematological tumors). Notably, vitamin B12 deficiency can cause macrocytic anemia and marked pancytopenia with high LDH values, often causing AIHA misdiagnosis [21]. Regarding hyperbilirubinemia, it is worth considering a possible concomitant Gilbert syndrome as well as the presence of an associated liver disease (in this case both indirect and direct bilirubin are increased). Haptoglobin, a scavenger of free hemoglobin, is the most sensitive among all hemolytic markers, being the last normalized after remission. Haptoglobin may be falsely low in the case of severe liver disease (as it is synthesized by the liver), and in the rare forms of familial hypoaptoglobinemia. In contrast, it increases during inflammation, nephrotic syndrome or smoking, thus masking an underlying hemolysis. Finally, ferritin may increase in chronic AIHA forms, and can be further increased by the concomitant presence of chronic inflammatory diseases, dysmetabolism, or familial hemochromatosis [21].

## 11. Clinical Signs and Symptoms in AIHA

On physical examination, common findings are pallor, cutaneous-mucosal jaundice, splenomegaly (in about half of cases), and hepatomegaly (1/3 of cases). Profound fatigue, dyspnoea, hypotension, tachycardia, and a soft eject systolic murmur may be present depending on the severity and acuteness of anemia onset. In chronic/relapsing forms, gallbladder stones and iron overload (increased ferritin and saturation of transferrin) are also frequent. About 10–15% of AIHAs have a severe clinical presentation, particularly when concomitant thrombocytopenia is present with related bleeding (Evans syndrome) [17,18]. The clinical picture may be complicated by serious infections, particularly after splenectomy and in relapsed and multi-treated patients. Acute renal failure due to renal hypoperfusion and massive hemolysis with hemoglobinuiria is a dreadful complication. Finally, in severe forms dominated by intravascular hemolysis, thrombotic complications are observed, more frequently venous (pulmonary embolism, disseminated intravascular coagulation and splanchnic thrombosis), but also arterial (myocardial infarction and stroke) [17,18,19]. The role of positive anti-phospholipid antibodies is not clearly established, while previous splenectomy is a recognized risk factor. Some symptoms are specific of CAD such as acrocyanosis and vasomotor phenomena in the superficial microcirculation (hands, feet, ears, nose, etc.), mostly triggered by exposure to cold or infections. Variation of 1–2 g of hemoglobin between the cold and warm seasons, are also reported [16]. PCH is characterized by an acute, occasionally severe onset, with chills, fever, cramps, lower back and abdominal pain, vasomotor phenomena, and urticaria. Anemia, due to intravascular hemolysis, may be severe with associated hemoglobinuria. PCH has been described in the past in association with syphilis, and currently it is mainly observed in children following viral infections. Finally, the specific signs and symptoms of the underlying disease must always be considered in secondary forms.

## 12. Therapy of wAIHA

Corticosteroids represent the first line of therapy. Generally, 1–1.5 mg/kg/day of oral prednisone for 3–4 weeks is able to increase hemoglobin and control hemolysis in 70–85% of cases [3,4,5,6]. Steroids should then be gradually tapered and stopped over a period of about 4–6 months, closely checking blood count and hemolytic indices. In patients with particularly rapid hemolysis and very severe anemia, or complex cases such as Evans syndrome, intravenous methylprednisolone at a 100–200 mg/day for 10–14 days or 250 to 1000 mg/day for one to three days may be indicated. Side effects of steroids, usually related to the dose and duration of therapy, should be managed rather than quickly reducing/stopping treatment [50]. After achieving a complete response, the patient should be regularly followed since only about 1/3 of patients remained in long-term clinical remission. For patients unresponsive to first-line therapy, early relapsed, or requiring inacceptable high doses (more than 10–15 mg prednisone per day), second-line therapy is indicated. In this case, a diagnostic re-evaluation for a possible underlying disease is advisable, since secondary AIHA are more often steroid-refractory [4,6,51]. Moreover, in the case of AIHA secondary to lymphoproliferative diseases, current guidelines advise the introduction of a lymphoma directed therapy including chemoimmunotherapy or small molecules according to patient- and disease-specific characteristics, and considering potentially hemolytic side effects (avoid fludarabine single agent) [4,7,8].

Rituximab (anti-CD20) is among the most recent and promising therapeutic options, with responses in 80–90% of cases (half of them complete), and a median duration of approximately two years. Median time to response is 4–6 weeks following the first dose, although responses after 3–4 months are not uncommon. Responses were observed regardless of prior therapy, and re-treatment was equally effective. Predictors of response are younger age, shorter interval between diagnosis and treatment, and early administration as second-line therapy [11,15,17]. Consistently, rituximab was used in a recent prospective randomized trial as first-line treatment in combination with steroids and was found superior to steroid monotherapy [13,14]. Moreover, a prospective pilot study has shown that first-line treatment with low-dose (100 mg weekly × 4) plus a short course of steroids compared favorably with conventional doses, with an overall response in 89% of cases (complete in 67%), and a relapse-free survival at three years in 68% of patients [11,52]. The drug has a well-established safety profile (infectious events in roughly 7%), although rare cases of progressive multifocal encephalopathy, mostly in onco-hematologic conditions, hepatitis B reactivation, and other viral infections have been reported. To prevent hepatitis B reactivation, antiviral prophylaxis is now recommended [50].

Splenectomy is considered the most effective second-line treatment of warm AIHA with an early response rate in about 70–80%, and a presumed curative rate in 20% of cases [3,4,5,6]. Moreover, a fraction of patients who fail splenectomy may be managed with lower doses of corticosteroids than those required before surgery. This option is recommended in young patients and in females who wish to become pregnant. On the other hand, age older than 65–70 years, cardiopulmonary disorders, previous history or serious risk of thrombosis, hepatitis C, underlying immunodeficiency, lymphoproliferative, and systemic autoimmune conditions should be carefully considered before surgery. The drawbacks of splenectomy are the lack of reliable predictors of outcome, the associated surgical complications, and most importantly, infectious and thrombotic complications. The former occurs in 3–5% of cases and has a mortality rate up to 50%, even after the recommended introduction of pre-operative vaccination against pneumococci, meningococci, and hemophilus [53,54,55]. Further recommendations include antibiotic therapy for three years post-splenectomy [1,56], annual flu vaccine, and revaccination against pneumococci and meningococci every five years [57]. Most importantly, patients should be educated to early administration of oral antibiotic and prompt referral to hospital in case of fever and suspected serious infection. Laparoscopic splenectomy is now preferred due to less trauma, fewer complications, shorter hospital stays, and an overall lower cost, but it should be performed by experienced operators. Thrombotic complications occur in 10–15% of AIHA, most frequently in severe cases with intravascular hemolysis and previous splenectomy. At present, there is no agreement on the use of anticoagulant prophylactic therapy, except in those cases associated with antiphospholipid syndrome. Given all these negative aspects and the availability of new treatments, the rate of splenectomy is progressively decreasing in recent years (from 15–20% to less than 10%).

Conventional immunosuppressive drugs (such as azathioprine, cyclophosphamide, cyclosporine), although widely used in the clinical practice mainly as steroid-sparing agents, are definitely moving to third line [3,4,5,6]. Response rates (mostly partial responses) are reported in 40–60%, but partially attributable to concomitant steroid administration [1,17]. As an exception, it is worth mentioning mycophenolate mofetil, which has been proven to be highly effective, particularly in children [58]. Among other treatments, danazol has been reported as effective in about 40% of cases in older studies [59,60] and high dose cyclophosphamide (50 mg/kg/day for four days) has shown responses in small series but with high toxicity [4]. Intravenous gammaglobulins have been used mainly for their low toxicity and for AIHAs secondary to infections, with an overall response in 40% of patients, better in a pediatric setting (60%). Finally, EPO has been successfully used in patients with multi-refractory AIHA, and may be indicated particularly in the presence of reticulocytopenia [17,25,61]. Moreover, it was proven useful in patients with very severe presentation, being able to reduce transfusion need and avoid hemolysis related to overtransfusion [49]. Table 3 summarizes the main treatments, response rates, and pros/cons.

## 13. Therapy of Cold Agglutinin Disease (CAD)

Treatment of CAD is primarily based on protection from cold, particularly of exposed parts of the body, and avoidance of cold infusions, food, and beverages. This measure is usually enough to control anemia (and to a lesser extent circulatory symptoms) in the less severe forms [4,16,62]. Surgery in hypothermia and/or cardiopulmonary bypass in CAD are a challenge and should be performed under normothermia [69,70]. Bacterial infection should be promptly treated. In critical situations, plasmapheresis daily or every second day is a temporary therapeutic option to be used together with specific therapy [71]. Notably, cold forms respond to steroids to a much lesser extent than warm AIHA, and generally require high and unacceptable steroid doses. Therefore, steroids may be used in the acute phase, but they should not be unnecessarily protracted for obvious side effects [4,16,72]. More importantly, splenectomy is ineffective in CAD and therefore it is contraindicated. This is not surprising as complement-mediated extravascular hemolysis mainly takes place in the liver and intravascular hemolysis in the bloodstream [16,72]. Today, CAD therapy is based on the early use of rituximab, effective as monotherapy in about 50% of cases. However, complete responses are rare (5–10%), and relapses are frequent (response duration of 1–2 years), albeit again responsive to re-treatment [9,73]. Combined treatment with rituximab and fludarabine orally (40 mg/m^2^ on days 1–5) resulted in higher response rates (76% of cases) and sustained remissions (estimated median response duration 6.5 years). However, hematological toxicities and infective complications were common, advising this regimen for refractory cases to 1–2 courses of rituximab [10]. Rituximab plus bendamustine combination therapy yielded a response in 71% of patients (40% complete) with long-lasting remission and an acceptable safety profile (11% experienced infection with or without neutropenia). Thus, this regimen is now advised as the first-line in relatively fit patients who are severely affected by CAD [13]. Finally, a small prospective study showed that bortezomib monotherapy (one cycle) was effective in about 1/3 of patients with CAD [63]. More recently, several case reports have described the efficacy of bortezomib associated with dexamethasone, vincristine, rituximab, and cyclophosphamide [64,65,66,67,68]. Other treatments (chlorambucil, interferon-alpha, cladribine, and cyclophosphamide) have shown little efficacy, mainly in small and old studies [16]. Table 3 summarizes the main treatments, response rates, and pros/cons.

## 14. New Drugs in wAIHA and CAD

Among the new drugs, B cell directed therapies and complement inhibitors are the most advanced in clinical trials (Table 4). The former are based on the close association and the several common pathogenetic mechanisms between lymphoproliferative and autoimmune diseases [5,7]. Monoclonal antibodies include ofatumumab, alemtuzumab (alone or in association with cyclosporine) and daratumumab, which have been mainly used in secondary AIHA. The BTK inhibitor Ibrutinib (currently used/under investigation in several lymphoproliferative disorders) has been shown to be effective in a case of AIHA associated with CLL and mantle cell lymphoma. The BCL2 inhibitor venetoclax, indicated for second-line treatment of CLL with a 17p deletion, has a potential role for refractory AIHA. Other BCR inhibitors directed against PI3K signaling (parsaclisib, NCT03538041) are under investigation in AIHA and CAD [7]. Complement modulation is the most promising therapeutic tool for CAD. Some effect with the anti-C5 monoclonal antibody, eculizumab, has been reported in the past, mainly on transfusion avoidance. However, terminal complement inhibition has no effect on C-mediated extravascular hemolysis, prompting investigation on proximal inhibition of the complement cascade [74]. The monoclonal antibody anti-C1s sutimlimab (formerly TNT003/BIVV009) has been proven to have a meaningful effect on hemoglobin levels, hemolysis, and fatigue, both in a pilot study mostly in secondary CAD [75,76] and in a prospective trial in primary CAD (NCT03347422, NCT03347396). Other ongoing studies are exploring the safety/efficacy of the C3 inhibitor pegcetacoplan (formerly APL-2) in CAD and in wAIHA (NCT03226678). Sirolimus has been used in secondary AIHAs and Evans syndrome, particularly in the pediatric setting [77,78,79]. An innovative strategy is inhibiting the spleen tyrosine kinases; one of these new drugs, fostamatinib, which also inhibits the B-cell receptor downstream pathway, has proven effective in various autoimmune diseases and is now in Phase 3 studies in wAIHA (NCT02612558). Finally, the safety/efficacy of several inhibitors of the neonatal Fc receptor (FcRn) such as orilanolimab (formerly SYNT001) are under investigation (NCT03075878). The FcRn is structurally homologous to the MHC Class I receptor family, is expressed by several cells, and is responsible for the salvage of IgG from catabolism. Blocking FcRn is a fascinating new strategy, which induces an increased clearance of IgG including that of pathogenic IgG autoantibodies.

## 15. Transfusion in AIHA

Red blood cell transfusion in patients with AIHA may be required to achieve and/or maintain clinically acceptable hemoglobin values until specific treatments become effective. The decision to transfuse should depend rather on the patient’s clinical status and comorbidities (particularly heart or pulmonary disease) than on the hemoglobin level. However, values lower than 6 g/dL generally deserve support [51]. Therefore, transfusions in AIHA should be deferred as much as possible, but should never be denied in cases of severe and clinically symptomatic anemia even though no truly compatible units can be found [3,62]. Pre-transfusion and compatibility tests are usually positive rising concerns in most clinicians, and transfused RBCs may have a reduced survival due to the presence of the autoantibody. Some authors recommend ignoring the specificity of the autoantibody and this indication has been demonstrated to be safe and effective in a great number of transfusions. Administration of steroids before transfusions is a matter of debate; this common practice (although unable to prevent truly incompatible transfusion reactions) has never been validated in controlled studies, but is not contraindicated in wAIHA as steroids are generally part of the standard therapy [1]. ABO- and RhD-matched red cell concentrates can anyway be safely administered in urgent cases if alloantibodies are reasonably excluded based on the previous transfusion and/or pregnancy history. In fact, alloantibodies are known to occur in about one third of AIHA patients, who are further at risk of alloimmunization for the particular state of activation of the immune system. These alloantibodies may be responsible for severe hemolytic reactions or may cause increased hemolysis that might falsely be attributable to an increase in the severity of AIHA. In non-urgent cases, alloantibodies should be excluded with absorption techniques or extended phenotypic typing (including KEL, JK, FY, and MNS systems) [80]. A recent review of the literature and survey of current practice showed a wide variability in the immunohematology testing algorithm and RBC selection: 98% used phenotyping, 80% performed genotyping at some point in the work-up, and 75% provided phenotype- or genotype-matched RBCs. The authors conclude that genotyping is the best technique that allows for the selection of antigen-matched units without laborious, costly, lengthy, and sample-consuming adsorption procedures [81]. Overall, a proper transfusion management of AIHA presumes a good communication between the clinician and the transfusion center: the first should promptly notify the need of a transfusion and its potential urgency; the second should carefully perform all pre-transfusion tests to ensure effective and safe transfusion. With regard to the volume to be transfused, it is worth reminding that overtransfusion (with an increased mass of RBCs available for destruction) should be avoided, particularly in elderly patients. Moreover, RBCs should also be administered slowly, possibly not exceeding 1 mL/kg/h. To minimize risks of febrile non-hemolytic reactions due to anti-leukocyte antibodies, leuko-depleted red cells are nowadays recommended in AIHA patients. Erythrocyte transfusions are generally more safe in CAD than in wAIHA, provided appropriate precautions, in particular, the patient and the extremity chosen for infusion should be kept warm, and the infusion should be particularly slow. Some authors recommend the use of an in-line blood warmer [62]. To avoid overtransfusion, rEPO administration may be useful and it has been recently reported to be effective in 70% of both primary and secondary AIHA patients with a median Hb increase of 2.4 mg/dL (range 2–83), independently of AIHA type and number of previous therapy lines [25].

## 16. AIHA in Solid Organ and Hematopoietic Stem Cells Transplant

Immune-mediated hemolysis may be a complication of solid organ transplantation in about 10–15% of cases, mainly when the donor is group 0 and the recipient group A. The phenomenon is named “passenger lymphocyte syndrome” and is due to the production of antibodies by the donor’s lymphocytes passively transferred to the recipient [82]. Hemolysis usually begins two weeks after transplant, may be severe, but is generally temporary since autoantibody production stops with the disappearance of the donor lymphocytes. The risk and extent of hemolysis are proportional to the lymphocyte mass contained in the transplanted organ: minor in kidney transplantation, intermediate in liver, and high in heart and lung transplantation [83]. An emerging and more severe clinical entity is AIHA after hematopoietic stem cell transplant (HSCT), which may complicate up to 2–4% after a median of 3–10 months, with a high mortality and poor response to therapies. Risk factors include use of unrelated donor and HLA-mismatch, occurrence of graft-versus-host-disease, use of cord blood, age < 15 years, cytomegalovirus reactivation, alemtuzumab use, and non-malignant condition pre-HSCT [84].

## 17. AIHAs Associated with New Biological Anti-Cancer Therapies

Tumor cells activate immune checkpoints such as molecular programmed death receptor-1 (PD-1) and cytotoxic T lymphocyte-associated antigen 4 (CTLA-4) pathways. Checkpoint inhibitors (CPIs) reactivate T lymphocytes to recognize cancer cells by blocking CTLA-4 or PD-1, and are therefore effective in numerous types of cancer, but have several immune-related adverse effects [6,7]. A recent revision of the database of the Food and Drug Administration reported a total of 68 cases: AIHA is the most commonly reported hematologic adverse event, mostly wAIHA, and frequently with a fulminant course (80% fatal mainly due to multi-organ failure and delayed diagnosis). Forty-three cases developed after nivolumab, 13 after pembrolizumab, seven after ipilimumab, and five after atezolizumab. The risk appeared higher for PD-1 or PD-L1 targeting agents than CTLA-4 inhibitors. The underlying diseases were mainly melanoma (41%), non-small cell lung cancer (26%), and others including kidney cancer, Hodgkin’s lymphoma, or skin cancers. The median time to AIHA onset was 50 days, with some patients developing concurrent thrombocytopenia, endocrine abnormalities, and gastrointestinal adverse events (colitis or hepatitis) [85]. In another recent analysis of 14 cases of AIHA after CPIs, m a high proportion of DAT negativity (38%) and of severe anemia (median Hb 6.3 g/dL) was found. Moreover, 50% of cases relapsed after the first line and 14% became chronic [86].

## 18. Conclusions

AIHA is a greatly heterogeneous disease due to the several immunologic mechanisms involved in its pathogenesis (cellular and humoral immune effectors, complement, cytokines, bone marrow compensation), possibly causing a clinically complex and severe disease. Moreover, the type and extent of the immune dysregulation may be different in each patient and also change over time, often determining an unpredictable clinical course. Diagnosis is usually easy, but difficult cases may challenge the general physician or internist, particularly if negative to common immune-hematologic tests or associated with lymphoproliferative neoplasms, autoimmune diseases, immunodeficiencies, drugs, solid tumors, or transplants. Bone marrow evaluation is increasingly advised, particularly in relapsed/refractory wAIHA and in CAD, to better define its pivotal role in pathogenesis and consequently harness therapy. The definition of the AIHA type (warm or cold) is fundamental as therapy is quite different and is further becoming targeted with the understanding of the peculiar pathogenetic mechanisms of these two forms. New therapies, directed against antibody-producing B-lymphocytes/plasma cells, complement components, or several kinases are under active development and will offer increased therapeutic opportunities to treat (and hopefully cure) the disease.

## Figures and Tables

**Figure 1 jcm-09-03859-f001:**
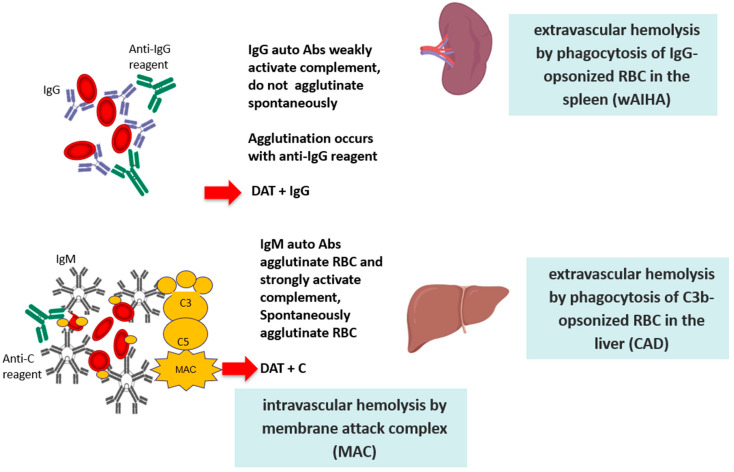
Pathogenic mechanisms of red blood cell (RBC) destruction.

**Figure 2 jcm-09-03859-f002:**
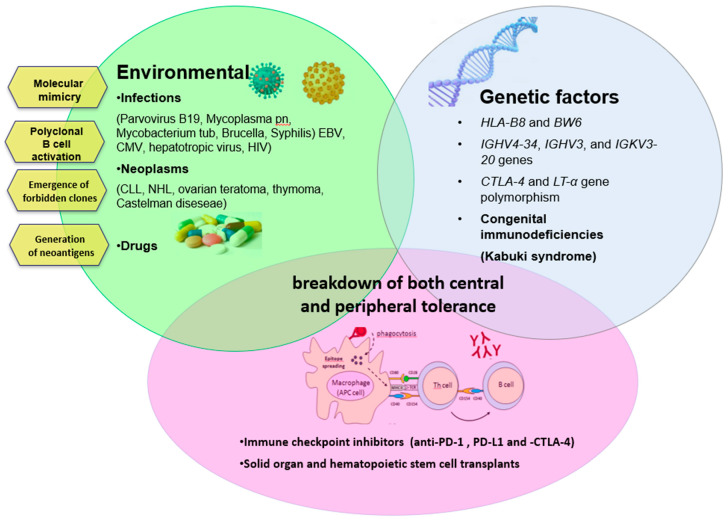
Immunologic, environmental, and genetic factors involved in the pathogenesis of autoimmune hemolytic anemia (AIHA). CLL: chronic lymphocytic leukemia, NHL: non-Hodgkin lymphoma, PD1/-L: programmed death 1 and its ligand, CTLA-4: cytotoxic T-lymphocyte-associated protein 4.

**Figure 3 jcm-09-03859-f003:**
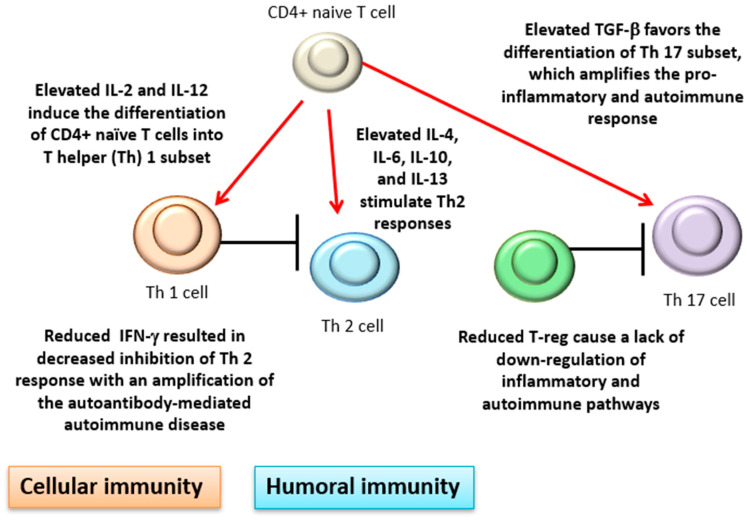
Cytokine dysregulation in AIHA. IL: interleukin; TGF-β: transforming growth factor β; IFN-γ: interferon γ. Red arrows indicate stimulation black lines inhibition/block.

**Table 1 jcm-09-03859-t001:** Autoimmune hemolytic anemia (AIHA): classification.

Autoantibody Characteristics				
	Class	Optimal T of Reaction (Range)	Specificity	DAT Positivity
Warm AIHA (wAIHA)	IgG (possible Complement fixation)	37 °C (0–40)	Rh system	IgG or IgG + C
Cold Agglutinin Disease (CAD)	IgM (common complement fixation)	4 °C (4–34)	I/i system	C
Mixed AIHA	warm IgG and cold IgM	4 °C and 37 °C	//	IgG + high titer cold IgM
Paroxysmal Cold Hemoglobinuria (PCH)	IgG (common complement fixation)	Reacts at 4 °C and hemolyzes at 37 °C	P Antigen	Positive Donath-Landsteiner Test

**Table 2 jcm-09-03859-t002:** Clinical and laboratory characteristics of patients at onset divided according to AIHA serological type.

	Median Age at Diagnosis (Years, Range)	Hb (g/dL), Median (Range)	LDH (ULN), Median (Range)	Ret (×10^9^/L), Median (Range)	Inadequate Reticulocytosis, *n* of pts (%)
wAIHA, IgG (158)	67 (5–94)	7.3 (2.1–14.1)	1.7 (0.6–26.7)	180 (22–644)	86 (54)
wAIHA, IgG + C (*n* = 67)	65 (21–92)	6.5 (2.0–11.5)	1.8 (0.8–7.2)	143 (53–641)	35 (52)
CAD (*n* = 107)	70 (28–94)	8.2 (4.0–13.5)	1.4 (0.3–12.2)	123 (13–644)	69 (64)
Mixed AIHA (*n* = 24)	61 (20–86)	6.4 (4.3–10.7)	1.7 (0.6–9.8)	181 (45–576)	15 (62)
Atypical AIHA (*n* = 22)	45 (25–78)	6.6 (3.0–10.9)	2 (0.7–18.1)	195 (29–780)	14 (64)

wAIHA: warm autoimmune hemolytic anemia; CAD: cold agglutinin disease; IgG: DAT positive for IgG; IgG + C: DAT positive IgG + C; LDH (ULN): LDH is expressed as folds of upper limit of normal.

**Table 3 jcm-09-03859-t003:** Standard therapies for warm AIHA (wAIHA) and cold agglutinin disease (CAD).

Treatment	Response Rates %	Comments	Ref
*wAIHA*			
Steroids	75–80	Curative in 20–30% only; short-term side effects (mood swings, psychosis); long-term side effects (diabetes, hypertension, infections, osteoporosis, cushingoid syndrome)	[1,3,4,5]
Rituximab	80–90	Effective even at low-dose (100 mg weekly × 4); re-treatment equally effective; well-established safety profile	[3,4,5]
Splenectomy	80	Potentially curative; unsuitable for elderly; surgical complications; thrombotic risk; life-long immune suppression; infection prophylaxis required	[3,4,5]
Azathioprine	40–60	Steroid-sparing agent; myelosuppression; infections; secondary malignancy; hepatic toxicity	[1,4]
Cyclophospamide	40–60	Steroid-sparing; myelosuppression; infections; urotoxicity; secondary malignancy; fertility problems/teratogenic	[1,4]
Cyclosporin A	40–60	Possible dose adjustment; hypertension; arrhythmia; myelosuppression; infections; nephrotoxicity	[1,3,4,5]
Mycophenolate Mofetil	80–100	Particularly effective in children; good safety profile; mild myelosuppression	[4,7,8]
Danazol	40	Steroid-sparing agent; low effectiveness in refractory cases; long-term hepatotoxicity	[4,8]
High dose cyclophosphamide	70	Response in small series; high toxicity (myelosuppression; infections; urotoxicity; secondary malignancy; infertility)	[3,4]
Intravenous immunoglobulin	40	Effective in AIHA secondary to infections; particularly effective in pediatric settings (60%); low toxicity	[4]
Recombinant erythropoietin	50	Effective in multi-refractory AIHA particularly with reticulocytopenia; useful in very severe presentation	[4,17,25]
*CAD*			
Steroids	15–30	Effective only at high dose; short-term side effects (mood swings, psychosis); long-term side effects (diabetes, hypertension, infections, osteoporosis, cushingoid syndrome)	[1,4,17,62]
Rituximab	50	Re-treatment equally effective; well-established safety profile	[9]
Rituximab/fludarabine	75	Sustained remissions (median response duration 6.5 years); hematologic and infectious toxicity mostly in older and frail patients	[10]
Rituximab/Bendamustine	70	Sustained remissions; infectious complications	[13]
Bortezomib	30	Side effects (neuropathy, neutropenia, thrombocytopenia, diarrhea, fatigue and rash)	[8,63,64,65,66,67,68]
Recombinant erythropoietin	50	Effective in multi-refractory cases particularly with reticulocytopenia	[4,25]

**Table 4 jcm-09-03859-t004:** New drugs in warm AIHA (wAIHA) and cold agglutinin disease (CAD).

	Mechanism	Route of Administration	Study Phase	Setting	Ref
B-cell directed monoclonal antibodies					
*Ofatumumab*	Anti-CD20	IV	Case report	Secondary AIHA	[4,7]
*Alemtuzumab*	Anti-CD52	SC	Case reports	Secondary AIHA	[4,7,8]
*Daratumumab*	Anti-CD38	IV	Case reports	wAIHA/CAD/Secondary AIHA	[4,7]
B-cell receptor inhibitors					
*Ibrutinib*	BTKi	Oral	Case reports	Secondary AIHA	[7,8]
*Venetoclax*	Bcl2	Oral	Case reports	Secondary AIHA	[4,7]
*Parsaclisib*	PI3Kδi	Oral	Phase 2	wAIHA/CAD	[4,7]
Complement inhibitors					
*Eculizumab*	C5i	IV	Case reports and phase 2	CAD/Mixed AIHA	[74]
*Sutimlimab*	Anti-C1s MoAb	IV	Phase1b and 3	CAD	[75,76]
*Pegcetacoplan*	C3/C3bi	SC	Phase1/2	CAD/wAIHA	[4,7]
T-cell directed therapies					
*Sirolimus*	mTORi	Oral	Case series	Evans’/Secondary AIHA	[77,78,79]
IgG-mediated phagocytosis inhibitors					
*Fostamatinib **	Syki	Oral	Phase 2	wAIHA	[4,7]
*Orilanolimab*	FcRn MoAb	IV	Phase1b	wAIHA	[4,7]

* Fostamatinib is also a B-cell receptor inhibitor; BTKi: bruton tyrosine kinase inhibitor; Bcl2: B-cell lymphoma 2; δPI3Ki: phosphatidylinositol 3-kinase-δ inhibitor; MoAb: monoclonal antibody; mTORi: mammalian target of Rapamycin inhibitor; Syki: Spleen tyrosine kinase; FcRn: neonatal Fc receptor.

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
