# Peer review of "New Insights in Autoimmune Hemolytic Anemia: From Pathogenesis to Therapy"

_jcm, 2020, doi:10.3390/jcm9123859_

Round 1

Reviewer 1 Report

This is a comprehensive and precise review article of autoimmune hemolytic anemia based on the experience of the authors that covers standard treatment to the topic of COVID-19.

There is no further issue to be concerned except for the minor points shown as below.

P1, line20-21

In the sentence of abstract part, In CAD~, it would be better to include ‘only’ as used in the table 3 to clearly show the ineffectiveness of steroid.

P6, line 212-213

Please check the grammar

reported reduced >>>reported to be reduced

P11, line 361

in ineffective>>>is ineffective

P10, table 3 and p12, table 4

The insertion of reference numbers for each treatment would be kind for the readers.

Author Response

We would like to thank the Referee for the positive feedback and for the careful revision. As suggested, we checked and corrected the mistakes throughout the manuscript and added reference numbers to tables 3 and 4

Reviewer 2 Report

This review is devoted to autoimmune hemolytic anemia, from pathogeny to treatment. It is divided in 16 paragraphs and 3 figures and 4 tables are included. It is well written, pleasant to read and easy to understand. It provides the expected information in a review dealing with AIHA.

I have nevertheless a few comments.

  1. In general, the distinction between primary and secondary AIHA is not evident, whereas both pathophysiology and treatment are different. For example, cold agglutinin disease (CAD) refers to cold antibodies secondary to any clonal lymphoproliferative disorder. Hemolytic anemia due to cold agglutinin should be referred to as Cold Agglutinin Syndrome (CAS) in any other case (infectious or other malignant disease for example) throughout the manuscript and may be differentiated in the treatment section.

  1. In the paragraph dealing with difficulties in AIHA diagnosis, the author could discuss false DAT-positivity in other hemolytic anemia and particularly mechanical hemolytic anemia and in anti-cardiolipin syndrome (line 103 to 111)

  1. The paragraph concerning genetic background in AIHA pathogenesis should include mention of ALPS (somatic or germline mutation in FAS, mutation in FASL, CASP10, CASP 8, NRAS or KRAS).

  1. The paragraph on immunologic mechanisms of AIHA is mainly dedicated to AIHA caused by therapies or pathogens. The authors could also deal with the mechanisms regarding AIHA secondary to immunodeficiency or lymphoproliferative disorders a bit further.

  1. Regarding cytokine dysregulations, the authors may add a world on the growing role of T follicular helper cells in self-tolerance break.

  1. Considering the paragraph on wAIHA, it mainly concerns primary AIHA, and this should be clarified. I fully agree that a diagnosis reevaluation for possible underlying disease is mandatory in patients refractory to steroids, but the authors should emphasize that in case of secondary AIHA, the treatment could differ. For example, diagnosis of monoclonal B lymphocytosis or CLL could raise the question of a specific treatment of the clone, especially in case of steroids refractoriness.

  1. In the abstract, in paragraph 13 and in table 4, why considering fostamatinib only as an IgG-mediated phagocytosis inhibitor ? Syk is also a key proximal kinase of the BCR pathway .

Author Response

We thank the Referee for the thorough revision and for the helpful suggestions.

I have nevertheless a few comments. 

  1. In general, the distinction between primary and secondary AIHA is not evident, whereas both pathophysiology and treatment are different. For example, cold agglutinin disease (CAD) refers to cold antibodies secondary to any clonal lymphoproliferative disorder. Hemolytic anemia due to cold agglutinin should be referred to as Cold Agglutinin Syndrome (CAS) in any other case (infectious or other malignant disease for example) throughout the manuscript and may be differentiated in the treatment section.

We agree with the Referee that the distinction between primary and secondary cases is important and could better delineated. We added the following sentences to address this issue, and also specified the differences between CAD and CAS. 

“Finally, AIHA may be classified as primary or secondary according to the absence/presence of underlying conditions. These include infections, lymphoproliferative syndromes, other autoimmune disorders, congenital immunodeficiencies, and others [4,6]. This distinction is important since may impact on therapy choice as discussed later. Regarding cold forms, the term CAD refers to cold antibodies secondary to any clonal lymphoproliferative disorder, whilst cold agglutinin secondary to infectious or other malignant disease should be referred to as Cold Agglutinin Syndrome (CAS) [4].”

  1. In the paragraph dealing with difficulties in AIHA diagnosis, the author could discuss false DAT-positivity in other hemolytic anemia and particularly mechanical hemolytic anemia and in anti-cardiolipin syndrome (line 103 to 111)

 As suggested, we added a sentence specifying the cases of falsely positive DAT test:

“Notably, the DAT may be falsely positive after the administration of various therapies (intravenous immunoglobulins, Rh immunoglobulins and anti-thymocyte globulins) and in diseases with elevated serum gammaglobulins or paraproteins, in the presence of IgG/IgM anti-cardiolipin antibodies and lupus anticoagulant, and seldom in mechanical hemolytic anemia.”

  1. The paragraph concerning genetic background in AIHA pathogenesis should include mention of ALPS (somatic or germline mutation in FAS, mutation in FASL, CASP10, CASP 8, NRAS or KRAS).

We agree with the Referee about the importance of the genetic landscape of AIHA associated to immunodeficiencies. The text has been modified as follows:

“Finally, the close association of autoimmunity and immunodeficiency is highlighted by two immunodeficiencies characterized by increased autoimmune phenomena: the autoimmune lymphoproliferative syndrome (ALPS) and the Kabuki syndrome (KS); the former is marked by organomegalies and by the presence of somatic or germline mutations of genes involved in apoptosis (i.e. FAS, FASL, CASP10, CASP 8, NRAS or KRAS); KS is marked by malformations and intellectual disability, and is caused by mutations in the KDM6A or KMT2D genes, involved in tolerance and immune system maturation [29,30].”

  1. The paragraph on immunologic mechanisms of AIHA is mainly dedicated to AIHA caused by therapies or pathogens. The authors could also deal with the mechanisms regarding AIHA secondary to immunodeficiency or lymphoproliferative disorders a bit further.

 As suggested, we added a discussion on the specific pathogenesis of AIHA secondary to lymphoproliferative syndromes and immunodeficiencies, both causes of immune disruption and emergence of the so called “forbidden clones”.

“More recent insights regard the production of autoantibodies in patients with CLL. CLL cells may facilitate autoimmunity by direct antigen presentation, and through the production of nonfunctional T regulatory cells that fuels the imbalance between Th17 cells and T-regs. Neoplastic cells may also produce autoantibodies in some cases, however, these are mainly polyclonal IgG produced by non-malignant self-reactive B-cells [2].”

“Finally, the basis of autoimmune phenomena is due to the break of “self-tolerance”, which is controlled by complex cellular mechanisms and cytokine patterns not yet fully understood [43]. This is the case of congenital immunodeficiencies, where autoimmunity arising from this complex interplay is also sustained by germinal mutations impairing tolerance maturation [7].”

  1. Regarding cytokine dysregulations, the authors may add a world on the growing role of T follicular helper cells in self-tolerance break.

The role of T follicular helper cells has now been included and the relative reference added:

“Follicular T-helper cells, that involved in the initiation and maintenance of immune responses that generate memory B cells and long-lived plasma cells, have been recently shown increased in autoimmune cytopenias and to contribute to autoantibodies production”

  1. Considering the paragraph on wAIHA, it mainly concerns primary AIHA, and this should be clarified. I fully agree that a diagnosis reevaluation for possible underlying disease is mandatory in patients refractory to steroids, but the authors should emphasize that in case of secondary AIHA, the treatment could differ. For example, diagnosis of monoclonal B lymphocytosis or CLL could raise the question of a specific treatment of the clone, especially in case of steroids refractoriness.

We agree with the Referee that wAIHA secondary to lymphoproliferative syndromes may deserve a distinct approach beyond first line steroids or in case of progression/treatment need of the underlying disease. The text has been modified as follows:

“Moreover, in the case of AIHA secondary to lymphoproliferative diseases, current guidelines advice the introduction of a lymphoma directed therapy, including chemoimmunotherapy or small molecules according to patient- and disease-specific characteristics, and considering potentially hemolytic side effects (avoid fludarabine single agent) [4,7,8].”

  1. In the abstract, in paragraph 13 and in table 4, why considering fostamatinib only as an IgG-mediated phagocytosis inhibitor ? Syk is also a key proximal kinase of the BCR pathway .

As a matter of fact, fostamatinib may also concur to dampen B cell activation and autoantibodies production. As suggested, we mentioned this mechanism within paragraph 13 and table 4.

“An innovative strategy is inhibiting the spleen tyrosine kinases; one of these new drugs, fostamatinib, that also inhibits B-cell receptor downstream pathway, has proven effective in various autoimmune diseases and is now in Phase 3 studies in wAIHA (NCT02612558).”
